# Time-Variant Seismic Fragility of Offshore Continuous Beam Bridges Based on Collapse Analysis

**Zhaodong Shi** 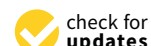**, Yan Liang \*, Yang Cao and Jialei Yan**

School of Civil Engineering, Zhengzhou University, Zhengzhou 450000, China; shizdlx@163.com (Z.S.); shicunfuyang@163.com (Y.C.); byronzhao@gs.zzu.edu.cn (J.Y.)

**\*** Correspondence: ylwan@gs.zzu.edu.cn; Tel.: +86-150-9306-2404

**Abstract:** In this paper, the concrete carbonation and chloride-induced corrosion of bridge structure in the service period under the offshore environment were comprehensively considered. Based on the time-varying degradation effect of mechanical properties of materials and continuous damage model, the time-varying seismic fragility of bridge components was analyzed with using incremental dynamic analysis. The time-varying brittleness curves of the bridge system and components were established according to the results of the analysis. According to the analysis of the time-varying fragility of the structure in the complete damage state, the collapse working conditions of the bridge structure and a method of quantifying the fragility coefficient were proposed. The results show that the fragility coefficient of the bridge system is higher than that of the components in the whole life cycle, and all of them increase with the increase of the bridge service cycle. When the peak acceleration of ground is small, the removing of $1^{\#}$ pier is more fragile. When reaching the design service life, the fragility coefficient of the bridge system is about 30% higher than that of the original state. The fragility coefficient of the bridge system in removing of $1^{\#}$ is the maximum value between three working conditions.

**Keywords:** bridge; durability damage; bridge seismic; collapse resistance; time-variant fragility

---

## 1. Introduction

Due to the influence of concrete carbonation and chloride-induced corrosion, the seismic performance of reinforced concrete bridge decreases continuously in the whole life cycle [1,2]. For the bridge located in the seismic belt, the durability damage will cause the change of dynamic characteristics and seismic response due to the decreasing of seismic performance and the change of bridge structure stiffness and damping, which increase the level of its damage state under earthquake. Hence, the reinforced concrete bridge that reaches the requirements of seismic design in the original state may not maintain safe operation in the earthquake disaster.

In recent years, the influence of durability damage on the seismic performance of concrete structure has been paid more and more attention. A large number of experiments have been carried out on the reinforced concrete components around the world. Concrete carbonation degrades the mechanical properties and surface alkalinity of concrete, resulting in cracking of concrete protective layer and corrosion of reinforcement [3–6]. Chen et al. [7] used XRD and ESEM to analyze concrete carbonation. The relationship between temperature, $CO_2$ concentration, relative humidity, and carbonation depth of concrete was obtained. Their relations are linear, power function, and polynomial, respectively. Based on diffusion-limited aggregation (DLA) model. He et al. [8] dynamically and deeply simulated the durability evolution process of the reinforced concrete structure after the coupling of chloride-induced corrosion and carbonation effects. Lee et al. [9] explored the influence of carbonation on the presence of water and acid-soluble chlorides, and the test results showed that the chloride-induced corrosion

was more obvious when it was combined with the carbonization process. However, it is more reasonable to use probabilistic analysis to evaluate the time-varying seismic performance of the bridge due to the uncertainty of structure and corrosion-related parameters [10]. As a probabilistic method, seismic fragility analysis has been widely used in the last few decades [11,12]. In recent years, some scholars have explored the time-variant seismic fragility of offshore bridges under chloride corrosion [13–15]. Choe et al. [16,17] proposed a probabilistic method of time-variant chloride-induced corrosion that is applied to reinforced concrete columns, and considered the reduction of their seismic performance. To identify the parameters which have the greatest influence on the fragility vulnerability of bridges, the sensitivity and importance analysis are carried out by using the nonlinear dynamic analysis. Padgett [18] established a time-variant fragility curve of multi-span continuous steel bridge by considering different seismic effects and corrosion parameters. Ghosh and Padgett [19] also considered the influence of reinforcement corrosion and steel bearing deterioration of bridge piers and established the time-variant fragility curve of the bridge. Dong et al. [20] proposed a framework by considering the impacts of structural deterioration associated with multiple hazards to assess the time-variant seismic performance of bridges. The authors considered the impact of flood scour, reinforcement bars corrosion and the concrete cover cracking on seismic fragility. However, these studies mainly discussed the time-variant seismic fragility of individual components. Nielson and DesRoches [21] analysed the influence of the main components on the fragility of bridge system based on the typical bridge, and utilized probabilistic method to directly assess the fragility of bridge system according to the individual component fragilities. The results show that the fragility of bridge system assessment by the first-order bounds could lead to errors of up to 40% when the fragility of bridge system is higher than that of any individual component. Based on the second-order bounds method, Liang et al. [22] assessed bridge system fragility, and the results showed that the first-order bounds is not as accurate as the second-order boundary method the second-order bounds method.

At present, the component fragility research has been relatively sufficient, and few studies on the system fragility have considered the coupling effect of durability damage, intensity measure and continuous damage of the bridge. In order to systematically analyze the service performance and maintenance value of the remaining system of bridge structures, a four-span offshore continuous beam bridge with rigid piers is selected for a case study. This paper adopts performance-based seismic design and fragility analysis method [23], considering the structure within the whole life cycle of concrete carbonation and chloride-induced corrosion. By using the exceeding probability of each component and system of the bridge under the action of an earthquake, propose the quantitative method of fragility coefficient. Moreover, based on this method, the fragile component and collapse resistance condition of the bridge is obtained, which can provide a reference for seismic analysis, evaluation, and maintenance.

## 2. Durability Damage Analysis of Materials

### 2.1. Concrete Carbonation

Concrete carbonation refers to the reaction of $CO_2$ in the environment with the $Ca(OH)_2$ in the 83 cement paste. It leads to reduction of its alkalinity and it destroys the passivation film on the reinforcement bar surface and reinforces bar corrosion and concrete cracking. Finally, the durability damage is caused [3–5]. In this case, the carbonation rate ($\bar{S}$) is selected as the influence factor of concrete carbonation, and the effect of carbonation depth and section size are considered. As shown in Equations (1)–(5) [22].

$$\begin{cases} A_c = \sum_{i=1}^{n} L_i \cdot x_c \\ x_c = k \cdot \sqrt{t} \\ \bar{S} = A_c / A \end{cases} \tag{1}$$

$$\sigma_{cp} = \left(1 + 0.619 \cdot \overline{S}\right) \cdot \sigma_p \tag{2}$$

$$\varepsilon_{cp} = \left(1 - 0.106 \cdot \overline{S}\right) \cdot \varepsilon_p \tag{3}$$

$$\varepsilon_{cu} = \left(1 - 0.459 \cdot \overline{S}\right) \cdot \varepsilon_u \tag{4}$$

$$E_c = \left(1 + 0.503 \cdot \overline{S}\right) \cdot E \tag{5}$$

where $\overline{S}$ is the carbonation rate; $A_c$ is carbonised area in mm$^2$; $A$ is the sectional area in mm$^2$; $L_i$ refers to the each side length of pier that the unit is mm; $x_c$ refers to the depth of carbonation in mm; $k$ refers to the coefficient of carbonation; $t$ refers to the time of carbonation in year; $\sigma_{cp}$, $\varepsilon_{cp}$, $\varepsilon_{cu}$, and $E_c$ are the peak stress, peak strain, ultimate strain, and elastic modulus of carbonised concrete, respectively; $\sigma_p$, $\varepsilon_p$, $\varepsilon_u$, and $E$ are the peak stress, peak strain, ultimate strain, and elastic modulus of concrete before carbonization, respectively; $\mu$ is Poisson's ratio which is 0.2.

According to the Equations (1)–(5), the changes of various mechanical parameters of C40 and C50 concrete over time during their service period and their comparison are shown in Figure 1. The case's design service life is 100 years, to discuss the seismic performance of the case after the completion of its service period. Hence, the service performance of each component and bridge system is analysed when the service period reaches 150 years. Therefore, the whole life cycle is divided into 0, 30, 50, 70, 100, and 150 years in the subsequent analysis of time-variant seismic performance.

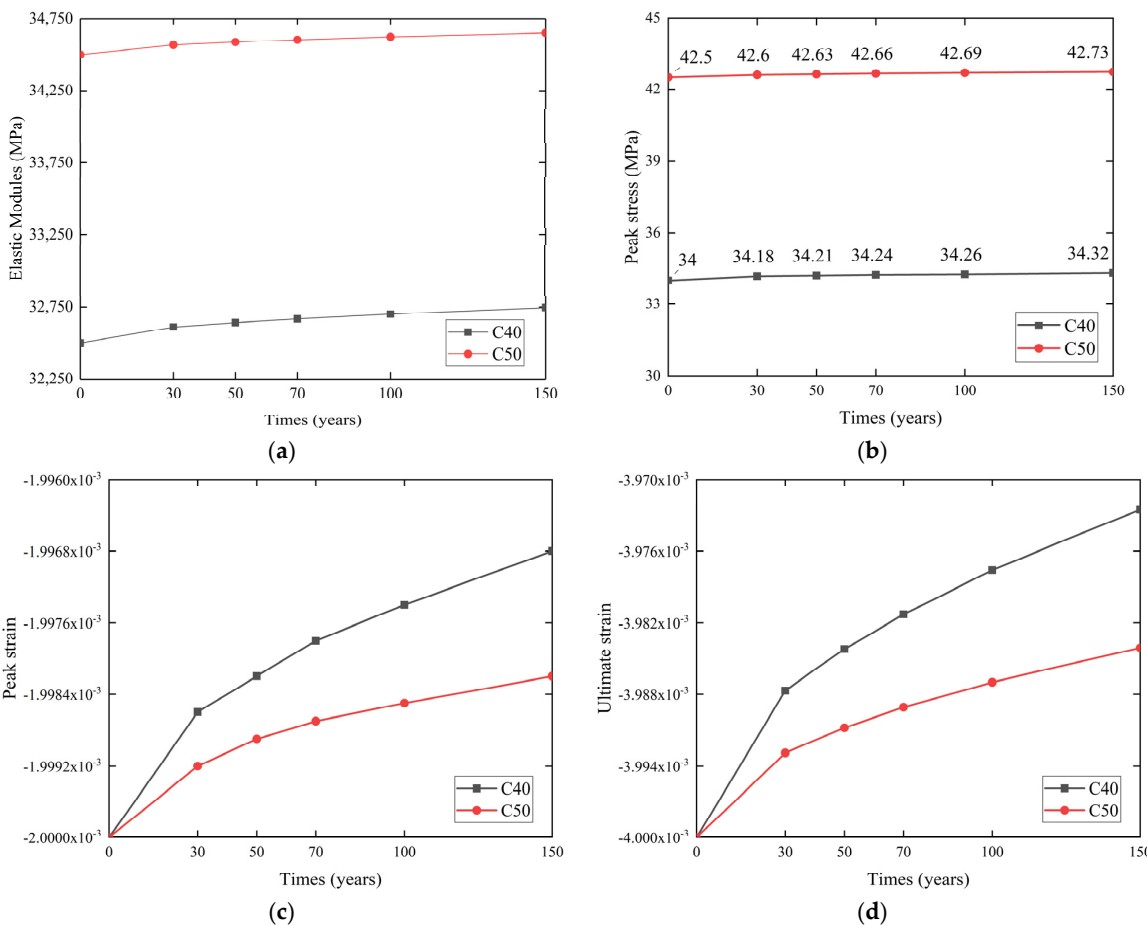

**Figure 1.** The mechanical parameters of concrete: (**a**) Elastic modulus; (**b**) Peak stress; (**c**) Peak strain; (**d**) Ultimate strain.

After considering carbonization effect, the peak stress, elastic modulus, and shear modulus of concrete all increase with the bridge service period, while the peak strain and limit strain both decrease

with the bridge service period increasing. Among them, C40 concrete changed more while C50 concrete changed less. Taking elastic modulus as an example, C40 and C50 concrete increased by 0.34%, 0.43%, 0.51%, and 0.61%; and 0.75%, 0.20%, 0.25%, 0.30%, 0.36%, and 0.44%, respectively, when they were in service at 30, 50, 70, 100, and 150 years compared with 0 years when they were in service.

## 2.2. Corrosion of Reinforcement Bar

Based on the "*Standard for durability assessment of concrete structures*" [24], the surface chloride ion concentration of concrete and the time of initial corrosion of steel reinforcement considering chloride-induced corrosion are calculated as shown in Equations (6)–(9) [15,17].

$$t_i' = \left(\frac{c}{K}\right)^2 \times 10^{-6} \tag{6}$$

$$K = 2\sqrt{D}erf^{-1}\left(1 - \frac{M_{cr}}{M_s}\right) \tag{7}$$

$$t_i = t_i' + 0.2t_1 \tag{8}$$

$$M_s = M_s' \cdot k \tag{9}$$

where $t_i'$ is the time when reinforcement bar begins to rust without considering chloride ion diffusion coefficient in year; $c$ is the concrete cover thickness in mm; $K$ is the chloride erosion coefficient in m$^2$ per year; $t_i$ is the time when the reinforcement bar begins to rust in the offshore environment in year; $t_1$ refers to the time of cumulative when the chloride reaches the stable value on the concrete surface in year, $t_1$ is 12.5 year in this case; $D$ is the diffusion coefficient of chloride ion in m$^2$ per year, $D$ is 12.5 m$^2$ per year in this case; $erf$ is the error function; $E_{rf}$ is the error function; $M_{cr}$ is the critical chloride ion concentration of reinforcement bar corrosion in kg per m$^3$; $M_s'$ is the concentration of chloride ion on the concrete surface at 0.1 km from the coast; $k$ is the correction coefficient of the distance from the coast.

Based on the "*Standard for durability assessment of concrete structures*" [24], and the research on the performance of reinforced concrete materials in the whole life cycle [25], yield strength, diameter deterioration, elastic modulus, and corrosion rate of reinforcement bar is shown in Equations (10)–(20) [24].

$$t_{cr} = t_i + t_c \tag{10}$$

$$t_c = \frac{\delta_{cr}}{\lambda_{cl}} \tag{11}$$

$$\delta_{cr} = 0.012c/d + 0.00084f_{cuk} + 0.018 \tag{12}$$

$$\lambda_{c1} = 0.0116i \tag{13}$$

$$\ln i = 8.167 + 0.618\ln M_{sl} - \frac{3034}{K} - 0.005\rho + \ln m_{cl} \tag{14}$$

$$M_{s1} = M_{s0} + (M_s - M_{s0})\left[1 - erf\left(\frac{c \times 10^{-3}}{2\sqrt{Dt_{cr}}}\right)\right] \tag{15}$$

$$\rho = k_\rho(1.8 - M_{cl}{}^u) + 10(RH - 1)^2 + 4 \tag{16}$$

$$\lambda_{cll} = (4.5 - \lambda_{cl}) \cdot \lambda_{cl} \tag{17}$$

When $\lambda_{cll} < 1.8\lambda_{cl}$, $\lambda_{cll} = 1.8\lambda_{cl}$

$$t_{cr} = \frac{0.602d_s\left(1 - 2 \times \frac{Xc}{d_s}\right)^{0.85}}{i_{corr}} \tag{18}$$

$$f_{yc} = (1 - 0.339\rho) \cdot f_y \tag{19}$$

$$E = (1 - 1.166\rho) \cdot E_s \tag{20}$$

where $t_{cr}$ refers to the time of concrete cracking, which unit is year; $t_c$ refers to the time of the reinforcement bar begins to rust until the concrete cracking in year; $\Delta_{cr}$ refers to the depth of reinforcement bar corrosion when the concrete is cracking, which unit mm; $\lambda_{cl}$ refers to the average annual rate of reinforcement bar corrosion before the concrete cracks, which unit is mm per year; $d$ refers to the diameter of reinforcement bar in mm; $f_{cuk}$ is the standard compressive strength of concrete cube in MPa; $i$ is the corrosion current density in μA per $cm^2$; $M_{sl}$ is the concentration of chloride ion on the reinforcement bar surface in kg per $m^3$. $M_{s0}$ is the chloride ion content added in concrete preparation in m3 per kg; $t_{cr}$ is concrete cracking time in year; $d$ is the diameter of reinforcement in mm; $c$ is the thickness of the concrete protective layer in mm. $K$ is atmospheric temperature; $\rho$ is concrete resistivity in KΩ· cm; $m_{cl}$ is the local environmental coefficient; $M_{s0}$ is the content of chloride ion mixed into concrete during preparation in kg per $m^3$; when the water–cement ratio is 0.3–0.4, or the concrete is C40 or C50, $k_\rho$ is 11.1; $M_{cl}{}^u$ is the average concentration of chloride ion on the concrete cover in kg per $m^3$; $RH$ is the environmental relative humidity; $\lambda_{cll}$ is the average annual corrosion rate of reinforcement bar after concrete cracking in mm per year; $X_c$ is the thickness of concrete protective layer in mm; $i_{corr}$ is corrosion current density of reinforcement in $cm^2$ per μA; $f_{yc}$ and $f_y$ are the yield strength of corroded and uncorroded reinforcement bars respectively; and $E$ and $E_s$ are the elastic modulus of corroded and uncorroded reinforcement bars, respectively.

The actual engineering has better performance due to the repair of concrete after cracking, the actual corrosion rate and material properties of the reinforced concrete structure have a large error with the calculated results [26]. To make the simulation results more accurate, the repair of the damaged concrete pier is considered in the finite element model in this paper. As shown in Equations (21)–(24) [26].

$$w = k(\Delta A_s - \Delta A_{s0}) \tag{21}$$

$$\Delta A_s = \frac{\pi}{4}\left(2\alpha \cdot x \cdot d_0 - \alpha^2 \cdot x^2\right) \tag{22}$$

$$x = 5.204\frac{(1 - w/c)^{-1.64}}{X_c} \cdot t^{0.71} \tag{23}$$

$$\Delta A_{s0} = A_s\left[1 - \left[1 - \frac{a}{d_0}\left(7.53 + 9.32\frac{x_c}{d_0}\right) \times 10^{-3}\right]^2\right] \tag{24}$$

where $w$ refers to the width of crack in mm; $k$ refers to the coefficient of calculation of the crack which is 0.0575; $\Delta A_s$ is the loss of reinforcement bar section in $mm^2$; $\alpha$ refers to the coefficient of corrosion, take 1 for uniform corrosion and 4 for non-uniform corrosion; $x$ is the corrosion depth of chloride ion in mm; $t$ refers to the bridge service period of the structure in year; $w/c$ refers to the ratio of water–cement, which is 0.5; $d_0$ refers to the diameter of reinforcement in mm; $\Delta A_{s0}$ is the loss of reinforcement bar section during cracking in $mm^2$; $A_s$ refers to the section place of reinforcement bar which is $mm^2$; and $X_c$ is the concrete cover thickness in mm.

Referring to the research of concrete crack conducted by Vu and Stewart [27], the protective layer is cracking while the crack width reached 1 mm, and the concrete of the bridge pier protective layer is repaired to reduce the corrosion rate of chloride ion to the value before concrete crack. According to Equations (21)–(24), the time distribution when concrete crack is 1 mm will be get, as shown in Table 1. According to Equations (10)–(20), the time-variant rules of corrosion rate, diameter, elastic modulus and yield strength of stirrup and longitudinal reinforcement in different concrete piers after repairing the concrete cover are shown in Figure 2.

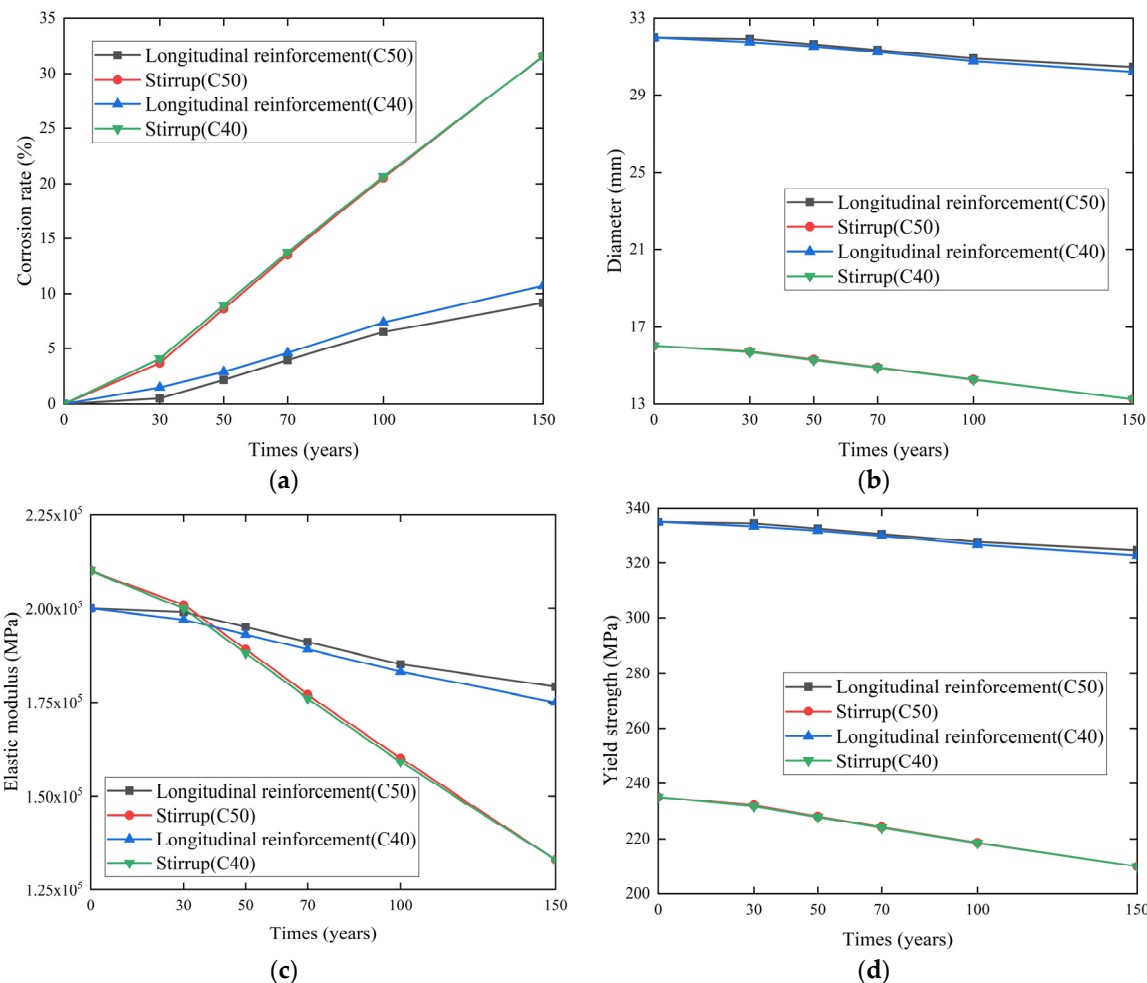

**Figure 2.** Longitudinal reinforcement and stirrup material properties: (**a**) Corrosion rate; (**b**) Diameter; (**c**) Elastic modulus; (**d**) Yield strength.

**Table 1.** Parameter analysis when the concrete surface crack reaches 1 mm.

| Steel Types | $X_c$ | $w/c$ | $\Delta A_{s0}$ | $\Delta A_s$ | $x$ | $t$ |
|---|---|---|---|---|---|---|
| Stirrup | 90 | 0.5 | $1.904\pi$ | $32\pi \times -4\pi \times 2$ | 0.240 | 1.495 |
| Longitudinal reinforcement | 106 | 0.5 | $2.452\pi$ | $64\pi \times -4\pi \times 2$ | 0.126 | 0.761 |

The yield strength, ultimate strength, elastic modulus and diameter of the reinforcement all deteriorate with the increase of the bridge's service life, and the damage degree of the stirrup is greater than that of the longitudinal reinforcement. Taking yield strength as an example, the yield strength of C40 and C50 concrete inner longitudinal reinforcement in 30, 50, 70, 100, and 150 years decreased by 0.51%, 0.98%, 1.56%, 2.52%, and 3.64%; and 0.17%, 0.74%, 1.34%, 2.22%, and 3.11%, respectively, compared with that in 0 years.

## 3. Fragility Analysis

### 3.1. Fragility Analysis Method

The fragility curve expresses the probability that the structure will exceed a specific limit damage state under a specific level of earthquake action, as shown in Equation (25).

$$P_f = P(R \geq LS | IM = x) \tag{25}$$

where $P_f$ is the exceeding probability; $R$ (Response) represents the seismic response of the structure or component; $LS$ (Limit State) represents a specific damage limit state; $IM$ (Intensity Measure) is the ground motion intensity parameter, namely the peak ground acceleration (PGA); $LS|IM$ notation in this equation means the specific failure state $LS$ of the structure at ground motion intensity $IM$; $x$ is the specific ground motion intensity level. Assume that the bearing capacity of the structure and the demand of seismic follow a log-normal distribution [28], as shown in Equation (26).

$$P_f[D \geq C|IM] = P_r\left(\frac{S_c}{S_d} \leq 1\right) = \Phi\left[\left(\frac{\ln \mu_D/\mu_C}{\beta}\right)\right] = \Phi\left[\left(\frac{\ln \mu_D/\mu_C}{\sqrt{\beta_D{}^2 + \beta_C{}^2}}\right)\right] \qquad (26)$$

where $D$ refers to the demand of seismic; $C$ refers to the bearing capacity of the structure; $IM$ refers to the parameter of ground motion intensity, namely, PGA [12]; $\mu_C$, $\beta_C$ and $\mu_D$, $\beta_D$ refers to the mean and logarithmic standard deviation of the seismic capability and seismic demand of the structure, respectively, according to HAZUS99 [29], when choosing PGA as the ground motion intensity parameter, $\sqrt{\beta_D{}^2 + \beta_C{}^2}$ is 0.5; $\Phi\,(\cdot)$ refers to the standard normal cumulative distribution function (CDF).

Shome et al. [30] and Cornell et al. [31] assume that $D$ and $IM$ are linearly correlated on a logarithmic scale, so $D$ can be expressed as a function of $IM$, as shown in Equation (27).

$$\ln(\mu_D) = \ln a + b \cdot \ln(IM) \qquad (27)$$

where $a$ and $b$ are coefficients determined by regression analysis of the nonlinear time-history analysis results. substitute Equation (27) into Equation (26) to obtain.

$$P_f[D \geq C|IM] = \Phi\left[2\ln\left(aIM^b/\mu\right)\right] \qquad (28)$$

### 3.2. System Fragility Analysis Method

The failure modes of each component in a bridge are not purely positive correlation or completely independent of each other, and the fragility of each component and its impact on the damage of the bridge system are also different. The error of the structural system fragility and the failure interval of the structure will be too large when the first-order bounds method is used, which affecting the seismic performance assessment of the structural system [21]. Consequently, it is more reasonable and practical to choose the second-order bounds method that considers the correlation in failure modes of components. The functional relationship is shown in Equation (29). The second-order bounds way is more sensitive to the exceeding probability. Hence, this paper ranks each component from large to small according to the exceeding probability [32,33].

$$P_{f1} + \sum_{i=2}^{n} max\left[P_{f1} - \sum_{j=1}^{i-1} P_{fij},\ 0\right] \leq P_{sys} \leq \sum_{i=2}^{n} P_{fi} - \sum_{i=2}^{n} max P_{fij} \qquad (29)$$

where $P_{sys}$ is the exceeding probability of the system, $P_{fi}$ and $P_{fj}$ are the exceeding probabilities of the $i$th and $j$th components respectively, $P_{fij}$ is the exceeding probability of the $i$th and $j$th components simultaneously, and $n$ is the number of possible destruction components.

### 3.3. Fragility Coefficient

The fragility coefficient and fragile component mentioned in this manuscript is the fragility coefficient of components and the most fragile component in a specific working condition. As the force condition of the structure will change in different working conditions, the fragility coefficient and fragile component will also change. Therefore, this paper proposes a quantisation method of fragility coefficient based on fragility curve of the complete damage of components. As shown in Equation (30).

$$Y_i = \frac{S_i}{a} \tag{30}$$

where $Y_i$ is the fragility coefficient; $S_i$ is the area enclosed by the fragility curve and the PGA axis when the $i$th component in the system reaches complete damage state. $a$ is the PGA value corresponding to the average exceeding probability of a component in the nonlinear finite element model when it reaches complete damage state.

## 4. Case Study

### 4.1. Bridge Description

A four-span offshore continuous beam bridge with rigid piers was selected as a case study to study the effect of the material deterioration on the seismic capacity of reinforced concrete bridge. The main girder section area is 9.52 m² with a single box and a single chamber, and C50 concrete is adopted. The torsional inertia moment of deck section $J$ is 25.76 m⁴. The second moment of area $I_z$ is 14.29 m⁴, and $I_y$ is 82.42 m⁴. 2#, 3# piers and 1#, 4#, 5# piers are made of C50 and C40 concrete, respectively. Longitudinal reinforcement and stirrup are HRB335 rebar that the diameter is 32 mm and R235 rebar that the diameter is 16 mm. The concrete cover thickness is 0.09 m. The bottom of the pier is a spread foundation, with C25 concrete. Seismic fortification intensity is 7 degrees, and the site class is in II. Further, the foundation is mostly weakly weathered rocks and the breeze rocks.

Taking the 2# rigid frame pier as an example, the pier section is hexagonal and its top is fixed with the main girder. The bottom is a 1.5 m thick spread footing foundation, and the concrete cover thickness is 9 cm, considering the confining effect of the core concrete by the stirrup, using Mander model cross section can be divided into confined and unconfined concrete, which is located in the core concrete and the concrete cover, respectively, as shown in Figure 3. In the finite element model, the pier body is divided into 37 nonlinear beam–column elements, and each element is provided with three integration points, except that the length of element 37 is 0.44 m. The other 36 elements are all 1 m. Fix the spread footing foundation at the bottom of the rigid frame pier without considering the interaction between pile and soil. The corner and displacement deformations caused by bond-slip are simulated by the zero length section element, the consolidation of the pier top and beam is simplified to a zero length element. The detailed data of box girder, bridge pier and support are shown in Figure 3 [34].

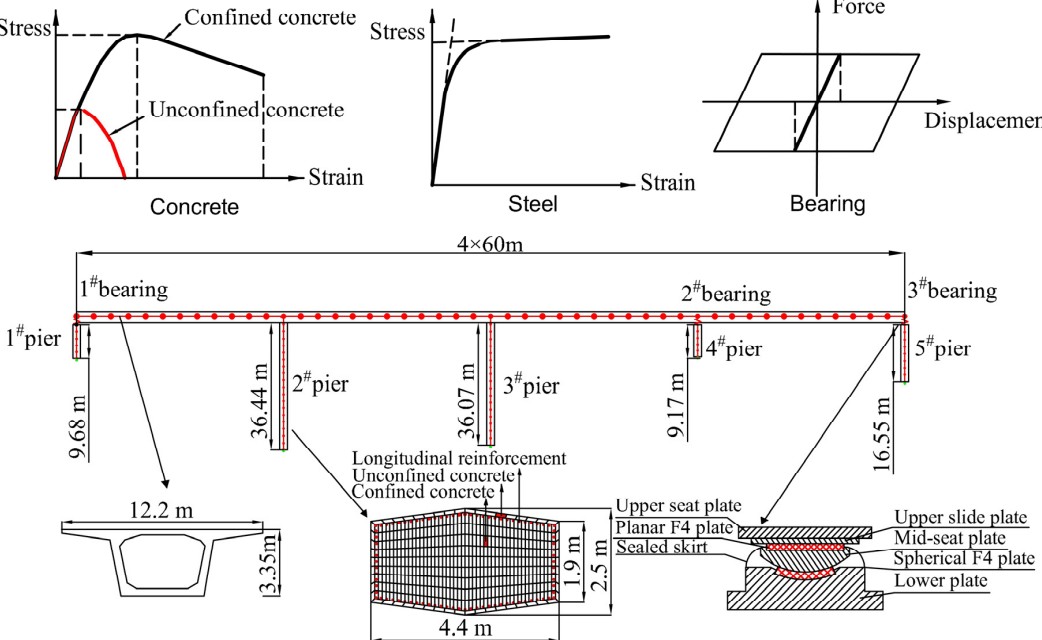

**Figure 3.** Finite element model and schematic diagram of the bridge.

### 4.2. Finite Element Model

The OpenSees platform is used to build a finite element model to simulate the bridge seismic response. In the seismic analysis, the concrete box girder as the superstructure does not occur plastic deformation. Therefore, the main girder in this paper is modeled using elastic beam–column element, and the element mass per unit length includes the self-weight and secondary loads. The plastic hinge region of the pier is considered to simulate its nonlinear characteristics better. Bottom of 1#, 4# and 5# piers set up plastic hinge area and top and bottom of 2# and 3# piers set up plastic hinge area. The pier is modeled using nonlinear beam–column element, and the whole pier element is divided into several fibre sections, as shown in Figure 3. Concrete01 in OpenSees is used to simulate the concrete behaviour, refer to the Mander et al. [35], to simulate the stirrup-confined concrete. The constitutive model of unconfined concrete and confined concrete are shown in Figure 3. Steel02 in OpenSees is used to simulate reinforcement bar, and its constitutive model is shown in Figure 3. Zero length section element are set at the bottom of the pier, and the strain penetration model of fully anchored steel reinforcement bars (Bond-SP01) is adopted to simulate the bond-slip effect in the plastic hinge area of each pier. $S_y$ is the yield slip, and is determined based on Zhao and Sritharan [36] as follows.

$$S_y = 2.54 \times \left[ \frac{d_b}{8437} \frac{f_y}{\sqrt{f_c'}} \times (2\alpha + 1) \right]^{1/\alpha} + 0.34 \tag{31}$$

where $d_b$ is the diameter of reinforcement bar, $f_y$ is the yielding strength of reinforcement bar, $f_c'$ is the concrete compressive strength, $\alpha = 0.4$ is the parameter of the local bond-slip relation CEB-FIP Model 90 [37], $S_u$ is the loaded-end slip when reinforcement bar reaches ultimate strength, and $S_u$ = 30~40 $S_y$. The top of 1#, 4# and 5# piers are installed the bearing, which is QZ12500 and QZ6000, respectively. This paper adopts the restoring force model of PTFE slide bearing (Figure 3) to carry out the simplified simulation. The sliding displacement calculation is shown in Equation (32).

$$x_y = \frac{f \cdot N}{K} \tag{32}$$

where $f$ refers to the coefficient of sliding friction, and the unit is taken as 0.02; $N$ refers to the load of superstructure; $x_y$ refers to the elastic displacement of the PTFE sliding bearing, and taken as 2 mm; $K$ refers to the stiffness of horizontal shear, the shear stiffness of QZ12500 bearing is 125,000 kN/m, and that of QZ6000 be is 60,000 kN/m.

### 4.3. Damage Index

The pier in this case study is easy to have bending failure that is the failure of ductile. Hence, the ratio of the displacement ductility is used to classify the damage state of the pier, which is expressed as Equation (33). According to the related study of Hwang et al. [11], damage index, as shown in Table 2.

$$\mu_d = \frac{\Delta}{\Delta_{cy1}} \tag{33}$$

where $\Delta$ refers to the relative displacement at the top site of pier, and $\Delta_{cy1}$ refers to the relative displacement of pier as soon as the longitudinal reinforcement reaches the first yield. The following four damage states can be obtained from the ratio of displacement ductility of the steel bar, the displacement ductility ratio at the equivalent yield of the section, the displacement ductility ratio at the compressive strain of concrete of 0.004 and the maximum displacement ductility ratio.

The relative displacements of 137 mm, 104 mm, 136 mm, and 187 mm are taken as the damage index of bearing refers to the damage index of relatively conservative, which derived from relative displacement and suitable for movable bearing brought up by Nielson [38].

**Table 2.** Damage index of piers.

| Pier Number | Slight Damage | Moderate Damage | Extensive Damage | Complete Damage |
|---|---|---|---|---|
| 1[#] pier | $1 < \mu_d \le 1.199$ | $1.199 < \mu_d \le 1.482$ | $1.482 < \mu_d \le 4.482$ | $4.482 < \mu_d$ |
| 2[#] pier | $1 < \mu_d \le 1.203$ | $1.203 < \mu_d \le 1.276$ | $1.276 < \mu_d \le 4.276$ | $4.276 < \mu_d$ |
| 3[#] pier | $1 < \mu_d \le 1.203$ | $1.203 < \mu_d \le 1.259$ | $1.259 < \mu_d \le 4.259$ | $4.259 < \mu_d$ |
| 4[#] pier | $1 < \mu_d \le 1.198$ | $1.198 < \mu_d \le 1.485$ | $1.485 < \mu_d \le 4.485$ | $4.485 < \mu_d$ |
| 5[#] pier | $1 < \mu_d \le 1.200$ | $1.200 < \mu_d \le 1.456$ | $1.456 < \mu_d \le 4.456$ | $4.456 < \mu_d$ |

### 4.4. Selection of Input Ground Motions

The target response spectrum is generated according to the bridge site condition, and 10 qualified seismic waves are selected from the PEER ground motion database based on the target response spectrum. Figure 4 shows that the mean value of the selected ground motion acceleration spectrum agrees well with the bridge design spectrum determined by "guidelines for seismic design of highway bridges" [39].

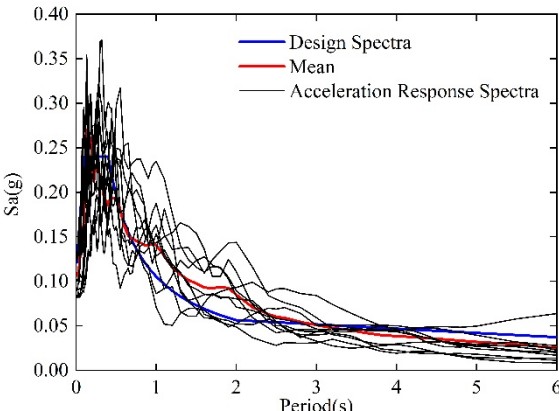

**Figure 4.** Acceleration spectra.

According to Equation (34), the PGA of 10 actual ground motion records are randomly generated into 10 groups of 150 ground motion records within the range of 0.01 g–1.0 g, and the PGA distribution is shown in Figure 5.

$$a_g^{(i)}(t) = k_i a(t) \tag{34}$$

where $a_g^{(i)}(t)$ refers to the ground motion record after the *i*th amplitude modulation; *a*(*t*) refers to the original ground motion record; $k_i$ refers to the amplitude modulation coefficient.

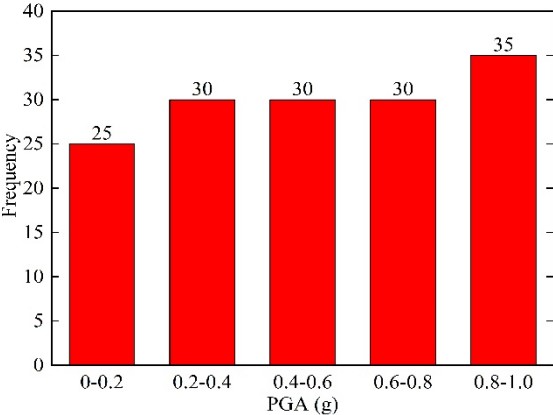

**Figure 5.** PGA distribution.

## 5. Time-Variant Fragility of Components and System

### 5.1. Time-Variant Fragility Analysis of Pier

Based on the above the article of Nielson [38], 150 ground motion and Equations (25)–(27), displacement ductility ratios of 150 seismic responses corresponding to each pier are selected for logarithmic linear regression analysis. Probabilistic seismic demand model (PSDM) as shown in Table 3 and Figure 6. Finally, the fitting functions are substituted into Equation (28) to obtain the regression coefficients of the 0-year probabilistic seismic demand model for each pier. $R^2$ refers to the degree of fitting in Table 3.

**Table 3.** The regression parameters and determination coefficients.

| Pier Number | ln $a$ | $b$ | $R^2$ |
|---|---|---|---|
| $1^{\#}$ pier | 1.517 | 1.273 | 0.601 |
| $2^{\#}$ pier | −1.010 | 1.021 | 0.602 |
| $3^{\#}$ pier | −0.934 | 1.015 | 0.600 |
| $4^{\#}$ pier | 1.446 | 1.372 | 0.585 |
| $5^{\#}$ pier | 0.568 | 1.136 | 0.612 |

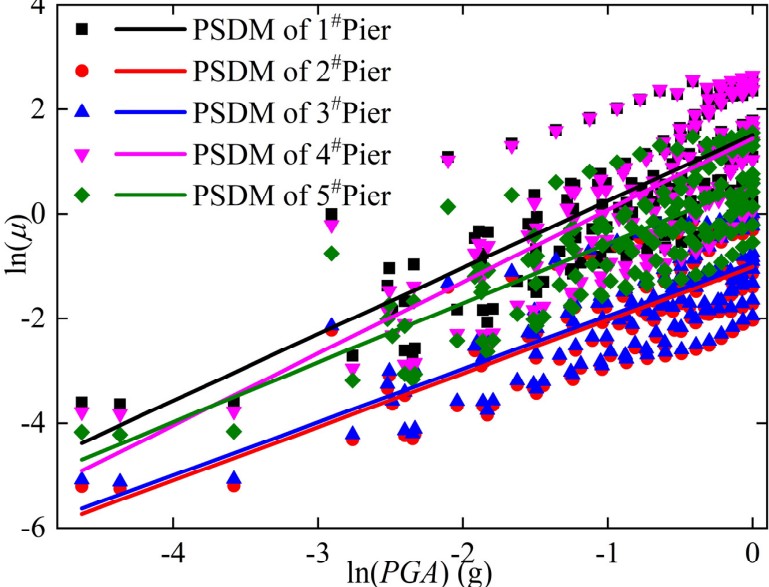

**Figure 6.** Probabilistic seismic demand model (PSDM).

Figure 6 shows that the seismic response of each pier increases linearly with the increase of PGA. The seismic response of $1^{\#}$, $4^{\#}$, $5^{\#}$ pier are greater than that of $2^{\#}$, $3^{\#}$ pier because $1^{\#}$, $4^{\#}$, $5^{\#}$ pier and $2^{\#}$, $3^{\#}$ pier are made of C40 concrete and C50 concrete respectively. In addition, $1^{\#}$, $4^{\#}$, $5^{\#}$ pier have bearing, while $2^{\#}$, $3^{\#}$ pier are rigid piers. During the deformation process under the action of earthquake, both the bottom and top of rigid pier will form a plastic hinge zone that dissipates seismic energy and the pier with bearing forms a plastic hinge zone at the bottom of pier, so the seismic response of rigid pier is less than that of the pier with bearing. In this paper, $1^{\#}$, $4^{\#}$ pier have similar height and same concrete type. They also connect with the main beam and similar bending resistance during the service period, as do $2^{\#}$, $3^{\#}$ pier. Moreover, the fragility coefficient in the following text only needs the fragility curve in the state of complete damage. Hence, only $1^{\#}$, $2^{\#}$ and $5^{\#}$ pier are represented to draw the time-variant fragility curve under the state of complete damage in the whole life cycle, as shown in Figure 7.

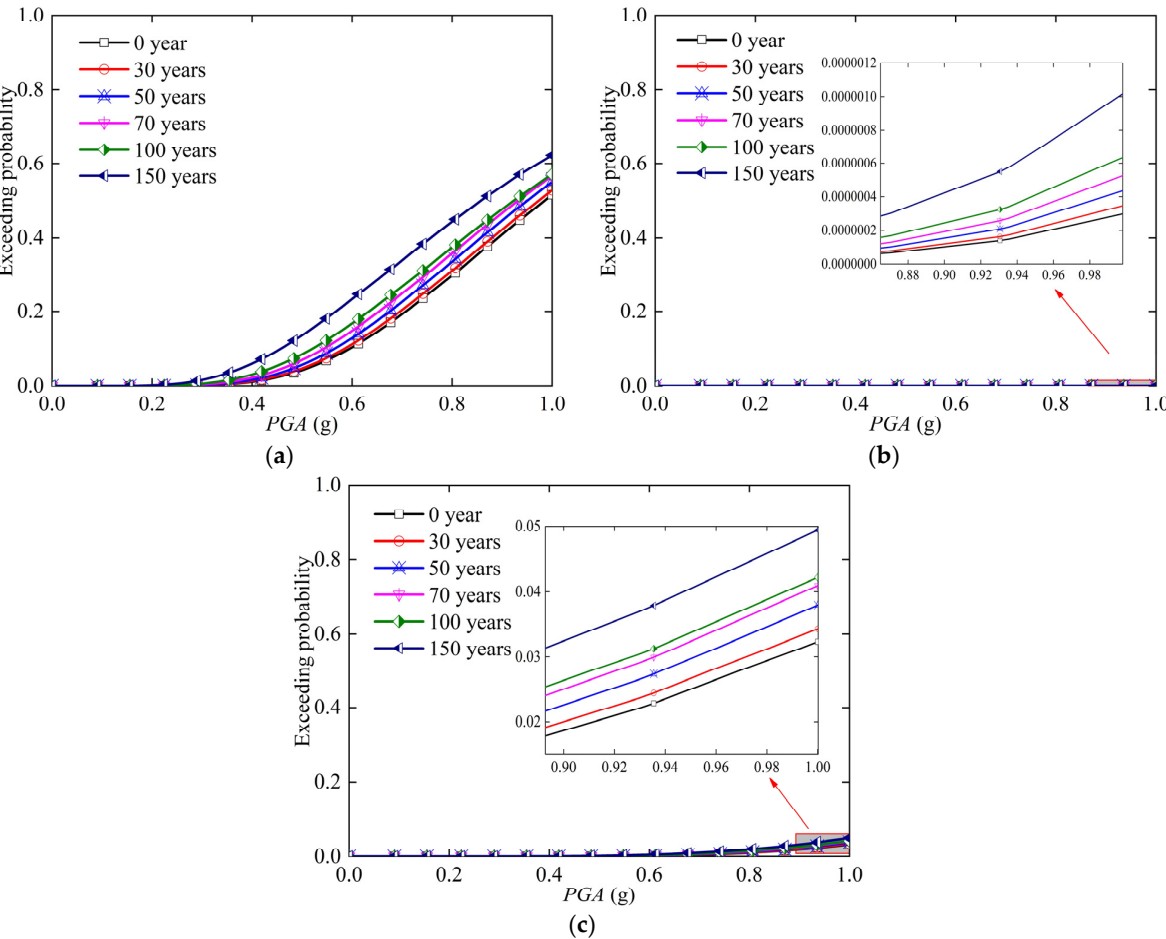

**Figure 7.** Time-variant fragility curve of piers in the state of complete damage: (**a**) 1# Pier; (**b**) 2# Pier; (**c**) 5# Pier.

Figure 7 indicates that the exceeding probability of pier in the whole life cycle is positively correlated with PGA and service period. In 0–30 years, the exceeding probability growth rate of each pier is the slowest because the cracking time of C40 concrete under chloride-induced corrosion is 27.1 years. The cracking time of C50 concrete under chloride-induced corrosion is 30.39 years, and it has almost no cracking in 0–30 years. When PGA = 1.0 g, the exceeding probability of 1# pier and 5# pier in 30, 50, 70, 100, and 150 years increased by 0.012, 0.035, 0.051, 0.055, 0.107 and 0.002, 0.006, 0.009, 0.01, 0.017, respectively, compared with 0 years. The exceeding probability increment of 1# pier is greater than that of 5# pier due to 5# pier is higher and the bending resistance is better than that of 1# pier. Figure 7b, c show that the exceeding probability in 150 years is $1.03 \times 10^{-6}$ and 0.049, respectively. Therefore, in the subsequent analysis, it can be assumed that 2# and 5# piers will not reach complete damage in the whole life cycle. In the same way, it is assumed that 3# rigid pier has similar height and materials with 2# pier will not reach complete damage in the whole life cycle.

In summary, considering that 2#, 3#, and 5# piers will not reach complete damage in the whole service period, the subsequent analysis of collapse resistance conditions is shown in Table 4 and Figure 8. Moreover, the subsequent analysis of collapse resistance performance of the bridge system should meet the following tow assumptions.

1. The working condition of "component dismantle" in the bridge system does not consider the additional load on the remaining new system.
2. When a component exceeds the damage index corresponding to its complete damage state, the component is considered to collapse.

**Table 4.** Working condition.

| Working Condition | Component Dismantle | Remaining Service Components of the System |
|---|---|---|
| WC1 | — | 1#, 2#, 3#, 4# and 5# pier;1#, 2# and 3# bearing |
| WC2 | Dismantle 1# pier | 2#, 3#, 4# and 5# pier; 2# and 3# bearing |
| WC3 | Dismantle 4# pier | 1#, 2# and 3# pier; 1# bearing |

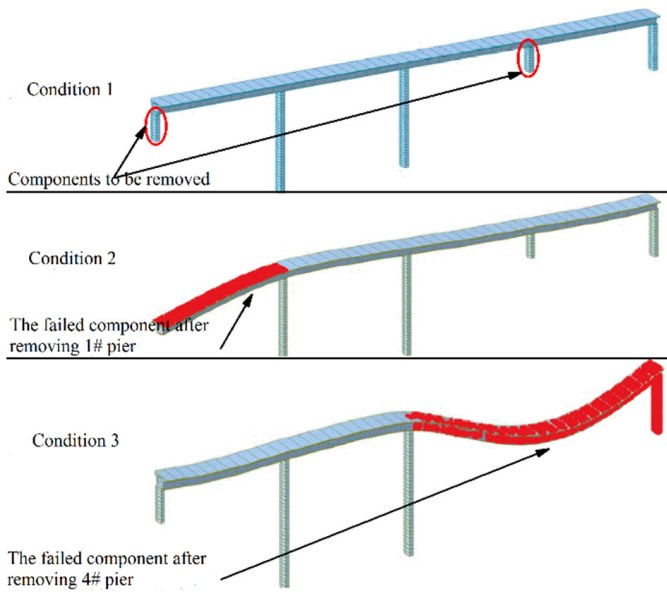

**Figure 8.** Collapse analysis.

### 5.2. Time-Variant Fragility Analysis of Bearing

Referring to the fragility analysis process of piers, only the time-variant fragility of the representative bearing in the state of complete damage is analyzed, as shown in Figure 9.

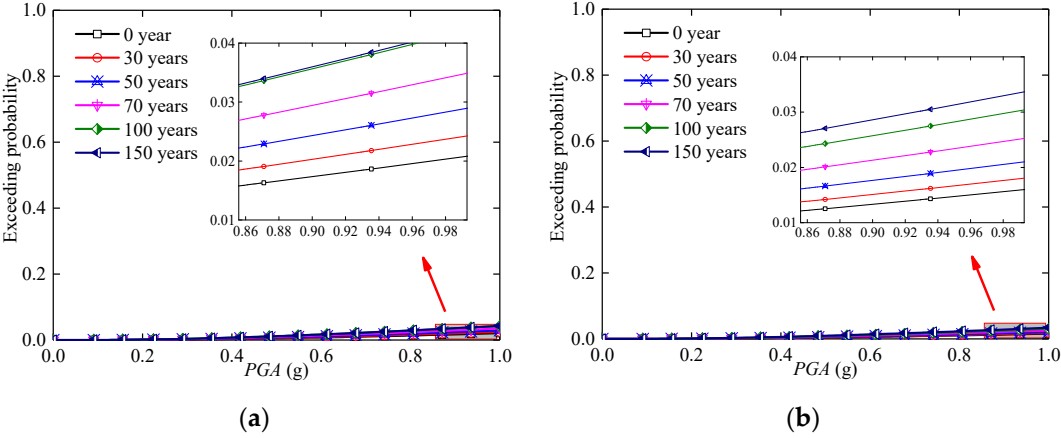

**Figure 9.** Time-variant fragility curve of bearing in the state of complete damage: (**a**) 1# Bearing; (**b**) 3# Bearing.

Figure 9 shows that the exceeding probability of bearing in the whole life cycle is positively correlated with PGA and service period. When it is time that PGA = 1.0 g, the exceeding probability of the bearing in the state of complete damage, as shown in Table 5. The exceeding probability of 1# bearing and 3# bearing in 30, 50, 70, 100, and 150 years increased by 0.00347, 0.00823, 0.01423, 0.0215, and 0.02191; and 0.00207, 0.00506, 0.00934, 0.01455, and 0.01784, respectively, compared with

0 years. The exceeding probability increment of 1[#] bearing is greater than that of 3[#] bearing because the seismic performance of the high pier with 3[#] bearing is better than that of the low pier with 1[#] bearing. According to the time-variant fragility curve under the state of complete damage of the two bearings, although the exceeding probability increased by 0.02191 and 0.01784 in 150 years compared with that of 0 years, the exceeding probabilities are only 0.04299 and 0.03405. Therefore, it is assumed that the bearing will not reach complete damage in the whole life cycle.

**Table 5.** Exceeding probability of bearing.

| Service Period (Years) | 0 | 30 | 50 | 70 | 100 | 150 |
|---|---|---|---|---|---|---|
| 1[#] Bearing | 0.02108 | 0.02455 | 0.02931 | 0.03531 | 0.04258 | 0.04299 |
| 3[#] Bearing | 0.01621 | 0.01828 | 0.02127 | 0.02555 | 0.03076 | 0.03405 |

### 5.3. Time-Variant Fragility Analysis of System

To discuss the difference between the exceeding probability of each component and the system, the fragility curves of each representative component and the bridge system in different damage states are compared, as shown in Figure 10.

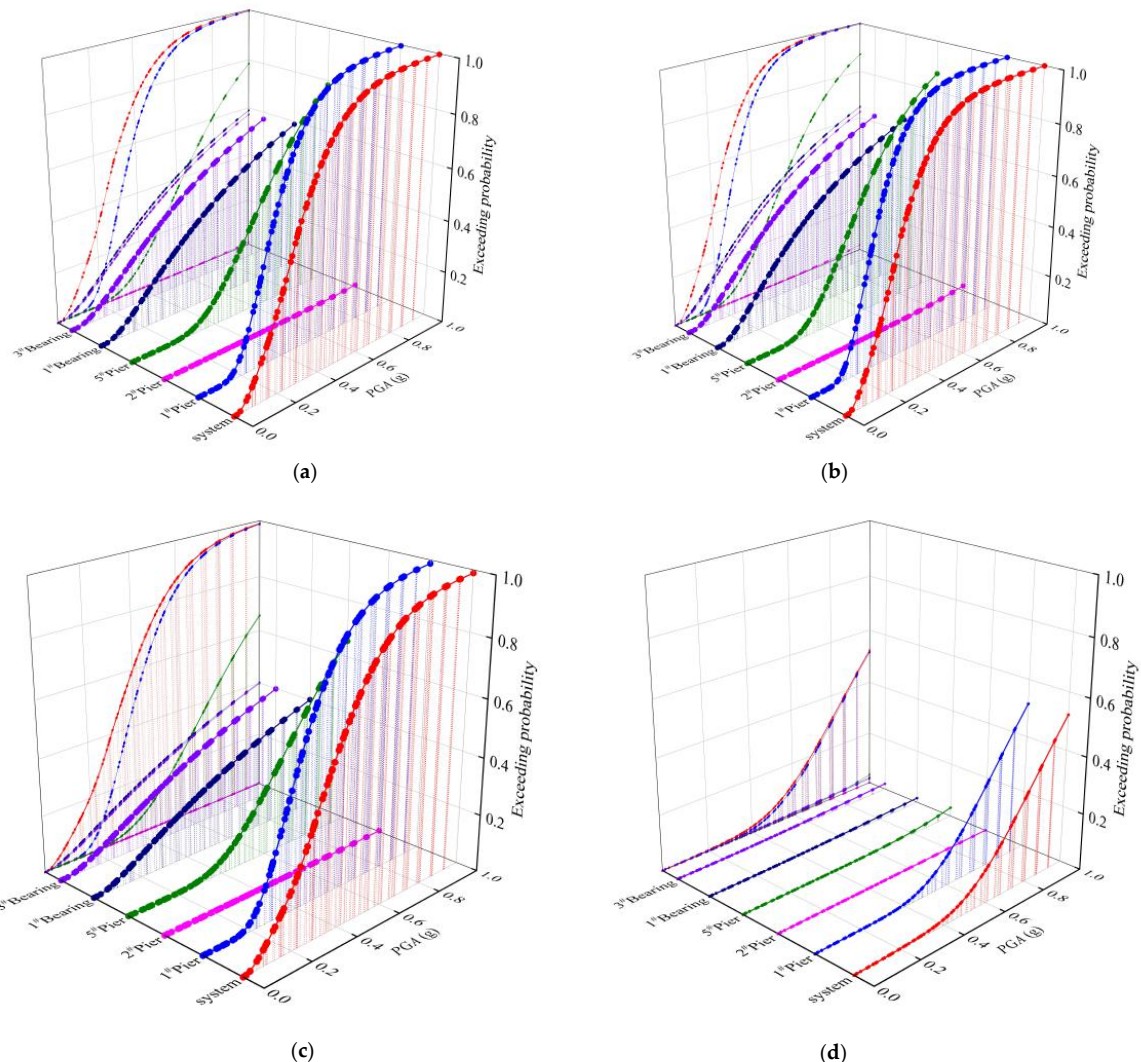

**Figure 10.** Fragility curves of each representative component and the bridge system: (**a**) Slight damage; (**b**) Moderate damage; (**c**) Extensive damage; (**d**) Complete damage.

Figure 10 shows that under the four damage states, the exceeding probability of the system is greater than that of each representative component. Therefore, under no circumstances can the fragility of a single component be used to assess the seismic performance of the bridge. Moreover, 2# pier is not discussed because it did not reach complete damage during the service period. However, for other components, in the slight damage (PGA < 0.575 g), moderate damage (PGA < 0.67 g), extensive damage (PGA < 0.69 g), and complete damage (PGA < 0.835 g), among all the components, exceeding probability of each representative bearing are greater than that of 5# pier. Further, when PGA in four states is greater than the above four values respectively, 3# bearing has the smallest exceeding probability, and the exceeding probability of each representative bearing is less than that of 5# pier. Moreover, the PGA value at the intersection of the fragility curve of 5# pier and representative component increases with the growth of service years. Therefore, the bridge system deformation is mainly borne by the bearing when the earthquake action is small, and the pier dissipates the vast majority of the earthquake energy when the earthquake action is large. With the increase of the service period, the energy dissipation effect of the pier in earthquake also increases gradually.

Ranking each component from large to small according to the exceeding probability, the bridge system's fragility surface in different states of damage in the whole life cycle is shown in Figure 11.

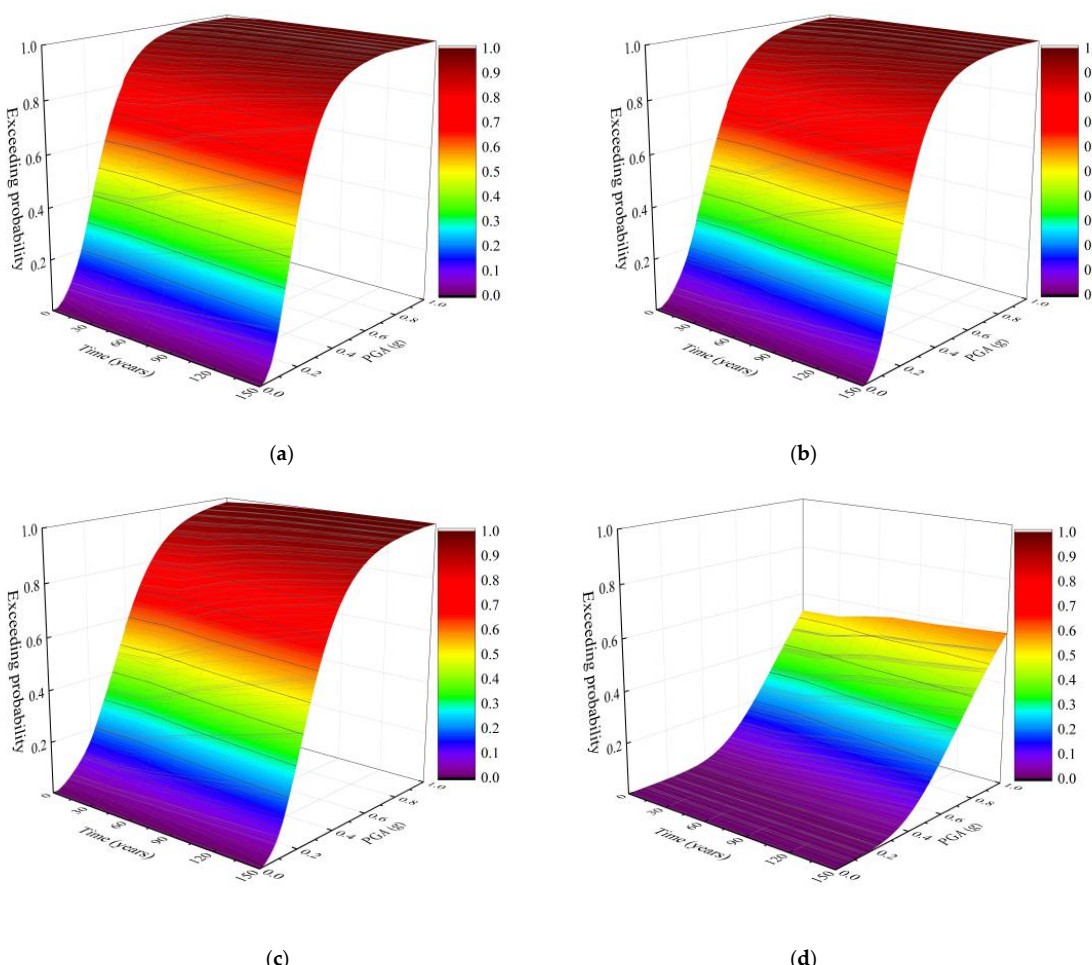

**Figure 11.** Fragility surface of the bridge system: (**a**) Slight damage; (**b**) Moderate damage; (**c**) Extensive damage; (**d**) Complete damage.

Figure 11 shows that the exceeding probability of the bridge system under different damage states is positively correlated with the PGA and service period. PGA = 0.5 g, for example, when the service period of the bridge reaches 30 years, its exceeding probability under four damage states increased by 1.01%, 1.11%, 1.12%, and 1.13%, respectively, compared with 0 years. When the service period of

the bridge reaches 100 years, its exceeding probability under four damage states increased by 4.20%, 7.62%, 13.60%, and 77.90%, respectively, compared with 0 years, which are significantly higher than the increment of 30 years. The increment increases with the increasing of PGA, this is because the pier begins to crack when its service period reaches around 30 years under the effect of chloride-induced corrosion. At this time, the seismic performance of the bridge structure is intact, and the increment of exceeding probability is small. However, when service period of the bridge reaches 100 years and each pier has been cracked for 70 years, the seismic performance of the bridge structure is poor and the increment of exceeding probability is large. Therefore, the time varying durability damage in service period have a great influence on the seismic ability of structures.

## 5.4. Time-Variant Fragility Coefficient of the System

To discuss the fragility of each component of the bridge system in the original state, the time-variant fragility curve and time-variant *a* value curve of each component and bridge system in the complete damage state of WC 1 in the whole life cycle as shown in Figure 12.

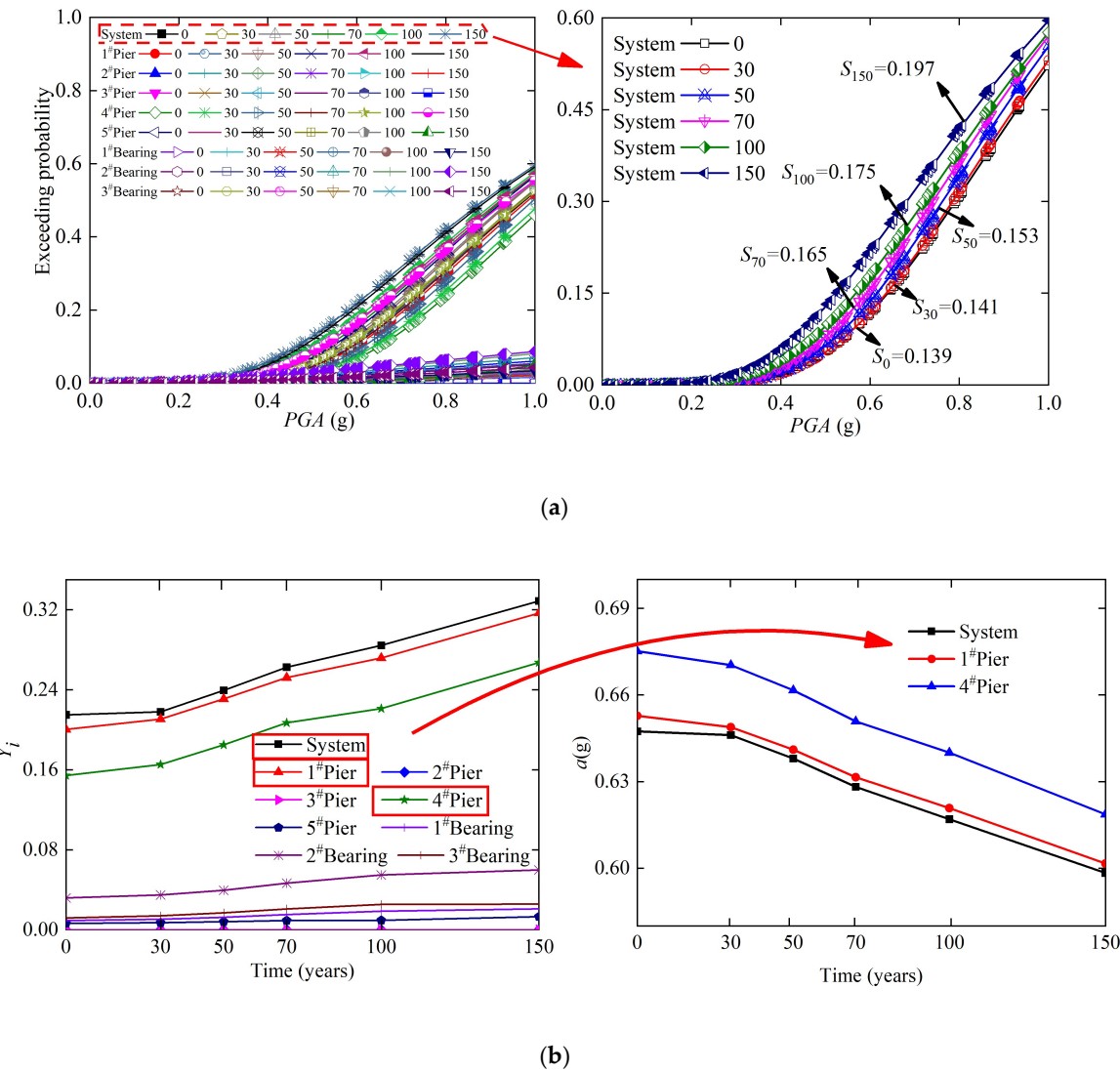

(**a**)

(**b**)

**Figure 12.** Fragility analysis in (working condition) WC1: (**a**) Time-variant fragility curve of each component and bridge system; (**b**) Time-variant fragility coefficient of components and bridge system.

Figure 12a shows that the exceeding probability of the bridge system under complete damage state is the highest in the whole working cycle. The area enclosed by the PGA axis and the curve of fragility,

namely $S$ increase with the extension of the service period. When the service period reaches 100 years, $S$ is 125.9% of that in 0 year. In addition to the 1# pier and 4# pier, the exceeding probability of other components in complete damage state is less than 0.1. When PGA = 0.5 g, the exceeding probability of each representative component (2#, 5# pier and 1#, 3# bearing) in complete damage state is that of 1# pier and 4# pier 0.00%, 0.00%, 0.98%, and 1.47%; and 9.88%, 14.82%, 13.45%, 20.18%, respectively. Compared with the 4# pier, the ratio of other components' exceeding probability to 1# pier is smaller that is more prone to occur complete damage during the service period. 2#, 3#, and 5# pier will rarely occur complete damage during the service period. Figure 12b shows that the fragility coefficient of components and the bridge system is positively correlated with service life. Moreover, the fragility coefficient of any component is lower than that of bridge system. $a$ value (the PGA value corresponding to the average exceeding probability) of any component is larger than that of bridge system. Hence, the bridge system is more prone to occur complete damage than any other component. During the service period, $a$ value of 1# pier is less than that of 4# pier. Taking the 100 years as an example, $a$ value of 4# pier is 3.1% more than that of 1# pier. Therefore, the most fragile component of the bridge system in WC1 is 1# pier.

According to Table 4 and Equation (30), the time-variant fragility coefficient of components and bridge system under different working conditions in the whole life cycle are shown in Figures 13 and 14.

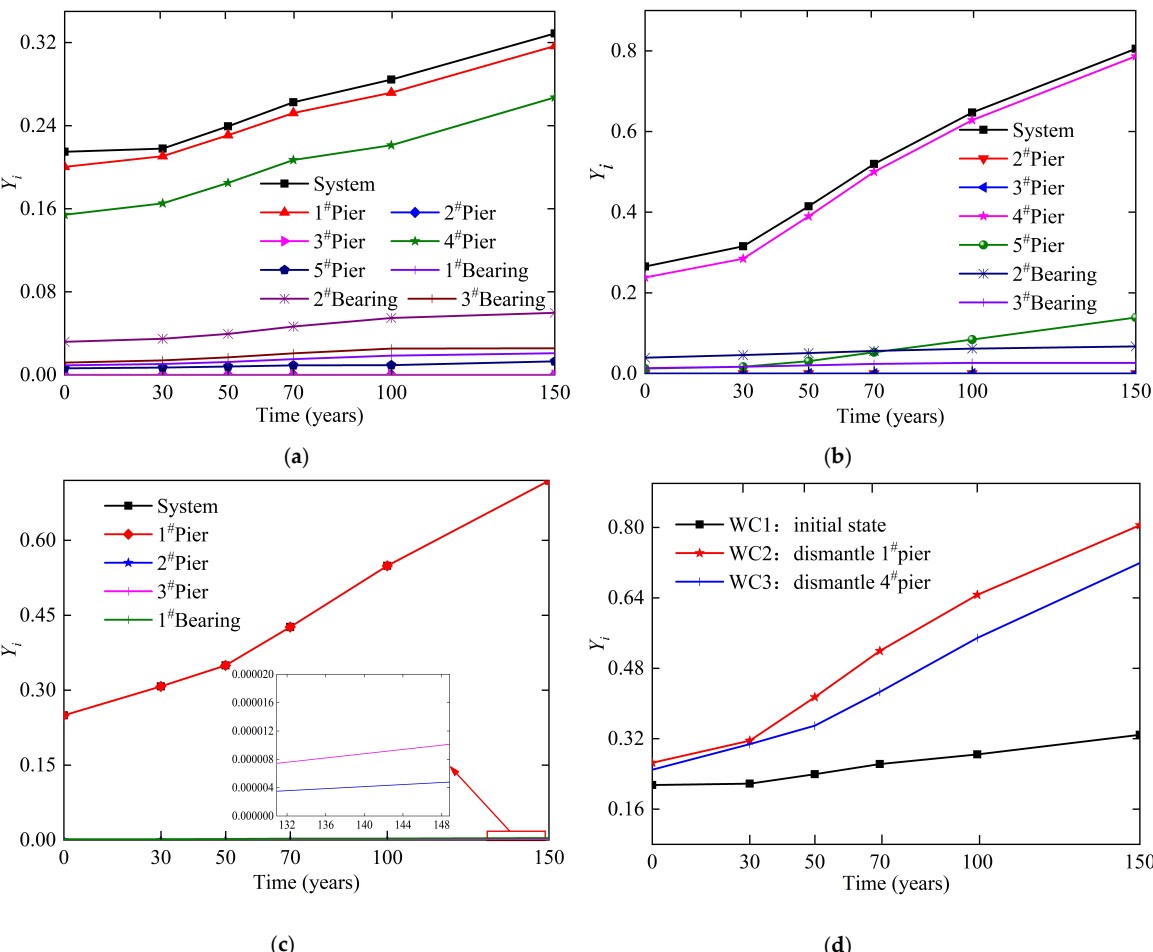

**Figure 13.** Time-variant fragility coefficient of each working condition: (**a**) WC1; (**b**) WC2; (**c**) WC3; (**d**) Fragility coefficient of the system.

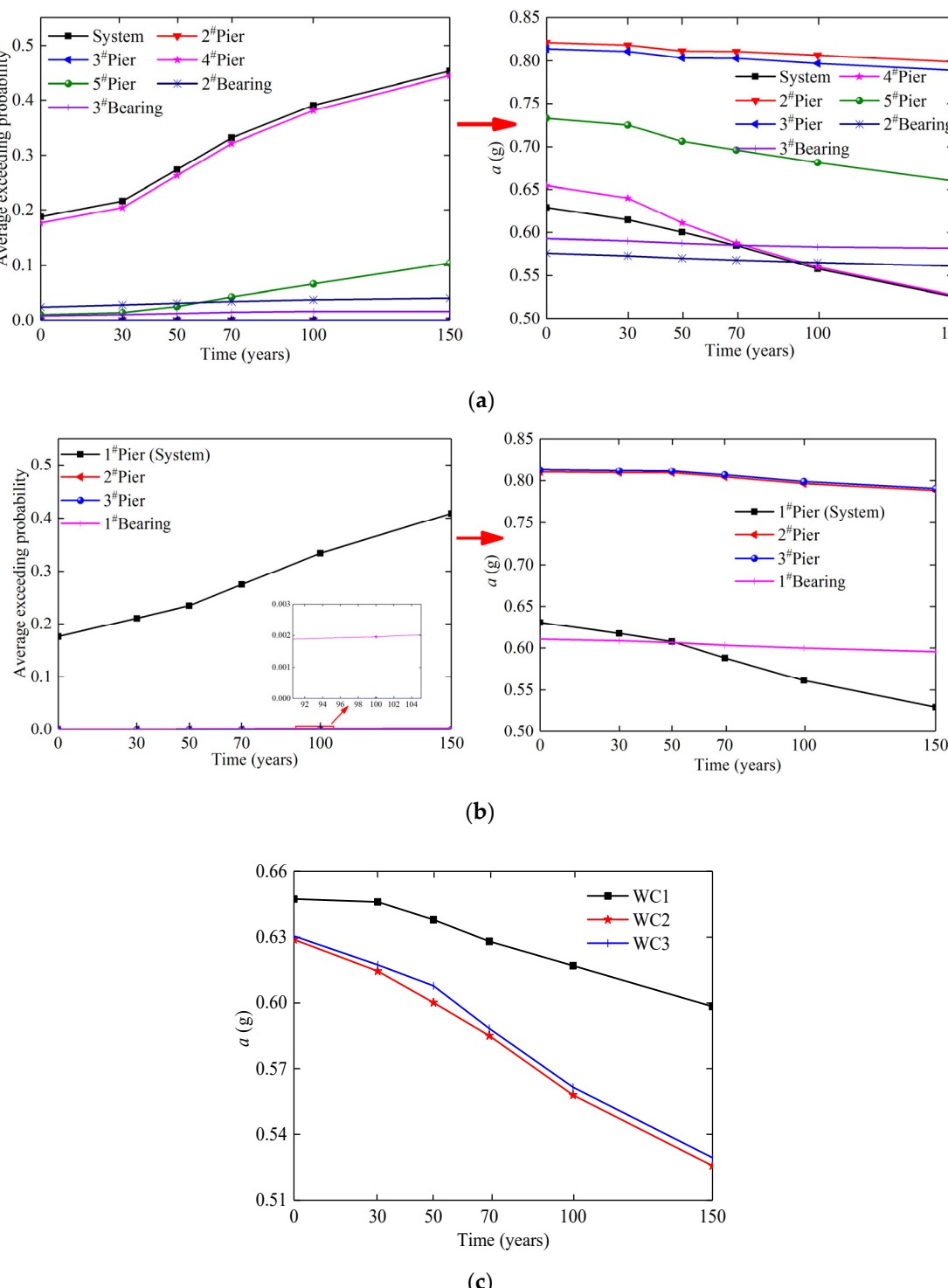

**Figure 14.** A value of each working condition: (**a**) Fragility analysis of WC2; (**b**) Fragility analysis of WC3; (**c**) Comparison of each working condition.

Figure 13a–c show the coefficient of fragility of every working condition promotes with the service period extension in the whole life cycle, and $4^{\#}$ and $1^{\#}$ pier with bearing are nearest to the bridge system. Taking 100 years as an example, the coefficient of fragility of the system is 105% in $1^{\#}$ pier and 132% of the bridge system in 0 year. Figure 13d shows that the fragility coefficient of WC2 and WC3 are greater than WC1 in the whole service period, and the fragility coefficient of WC2 is the largest. The fragility coefficient of WC2 in 100 years is 227.53% and 117.85% of WC1 and WC3, respectively. It is

because that compared with the WC3 which is formed after the complete damage of $4^{\#}$ pier, the system stiffness and redundancy of WC2 which is formed after the complete damage of $1^{\#}$ pier is less than WC3. The seismic ability of the remaining bridge system of WC3 is better than WC2. Additionally, the fragility coefficient of WC2 and WC3 are not significantly different in 0–30 years, but they were significantly different in 30–150 years. It is because that the C50 concrete pier can hardly get damaged in the service period, while C40 concrete pier becomes the key component of the seismic system. There are two C40 concrete piers in WC2, which are $4^{\#}$ pier and $5^{\#}$ pier, respectively. In contrast in WC3, only $1^{\#}$ pier is C40 concrete pier, and C40 concrete cracks in 27.1 years. Therefore, the durability damage of components in 0–30 years is small. The fragility of the two working conditions is not much different. With the extension of the service period, the features of low stiffness and the features of easy to durability damage in WC2 gradually become prominent. The fragility coefficient of WC2 starts to be significantly greater than that of WC3. Figure 14a,b shows that without considering the component whose average exceeding probability is nearly 0, the exceeding probability and *a* value of the bridge system in complete damage state are the maximum and minimum values of every working conditions in the service period. In addition, $4^{\#}$ pier and $1^{\#}$ pier are the most fragile components in WC2 and WC3, respectively. Figure 14c shows that the PGA value in WC2 and WC3 is less than that in WC1 when complete damage occurs, and the *a* value in WC 2 is the lowest. Taking the 100 years as an example, the *a* value of WC2 is 90.44% and 99.36% of WC1 and WC3, respectively. In summary, WC2 is the most fragile working condition in the service period of the bridge.

## 6. Conclusions

In this paper, a four-span offshore continuous beam bridge with rigid pier is selected for a case study. Based on the fragility analysis method and considering the influence factors such as concrete carbonation and chloride-induced corrosion, the time-variant fragility of the bridge is analyzed, and the quantification method in obtaining the coefficient of fragility is proposed. The main conclusions are as follows:

1. After considering the time-varying damage of material durability, the peak stress of concrete increases with the extension of service period. On the contrary, the peak strain and limit strain of concrete decrease with the extension of service period. Moreover, the yield strength, ultimate strength, and diameter of the reinforcement decreased with the extension of service period. In addition, the stirrup damage degree was worse than the longitudinal reinforcement. As C50 concrete has better material properties than C40 concrete, the longitudinal reinforcement damage degree in $2^{\#}$ pier and $3^{\#}$ pier is lower than that of $1^{\#}$, $4^{\#}$ and $5^{\#}$ piers. Taking $1^{\#}$ pier which is in 100 years as an example, the yield strength of longitudinal reinforcement and stirrup decreases by 2.52% and 7%, respectively, after the repair of concrete damage in the protective layer of the pier.

2. When the local vibration intensity is small, the $1^{\#}$ pier in working condition 1 is more fragile to damage at the component level. The bridge system in working condition 2 is more fragile at the system level in small local vibration intensity. According to the time-variant fragility curves of components and bridge system, the fragility of each component and bridge system increases with the extension of the service period. Meanwhile, the exceeding probability of any component is lower than the system in same condition. Therefore, the exceeding probability of component cannot be used to replace the exceeding probability of bridge systems when assessing the seismic performance of a bridge.

3. The time-varying fragility coefficient of each component and system in three working conditions are compared, among which the fragility coefficient of system are the largest values under every working conditions. When the bridge reaches the design service life, the fragility coefficient of the bridge system is 105% of $1^{\#}$ pier and 132% of the bridge system in 0 year. During the whole service period, both WC2 (dismantle $1^{\#}$ pier) and WC3 (dismantle $4^{\#}$ pier) will increase the fragility coefficient of the system compared with WC1 (original state). However, WC2 is the most fragile working condition in the service period of the bridge.

**Author Contributions:** Y.L. proposed the topic of this study and performed the finite element analysis; Z.S. designed the process and wrote the paper; J.Y. designed the experiments; Y.C. has done related works in review and editing. All authors have read and agreed to the published version of the manuscript.

**Funding:** This work was supported by the National Natural Science Foundation of China (Grant No. 51608488), and the Scientific and Technological Project of Henan province, China (192102210185).

**Conflicts of Interest:** We declare that we do not have any commercial or associative interest that represents a conflict of interest in connection with the work submitted.

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
