# Peer review of "Time-Variant Seismic Fragility of Offshore Continuous Beam Bridges Based on Collapse Analysis"

_applsci, doi:10.3390/app10238595_

Round 1
Reviewer 1 Report
In this manuscript, the effect of concrete carbonation and chloride-induced corrosion on the dynamic characteristics and seismic response for the bridge concrete structure were studied
The reviewer believes that
The manuscript needs to be improved with minor modifications.
Detailed comments are as follows:
1. Please discuss the results of figures 1, 2, 3 and 4
2. Please discuss the validation of the numerical models (finite element model) with more details.
Reviewer 2 Report
Comment to the Authors
Review of the paper “Time-variant Seismic Fragility of Offshore Continuous Beam Bridges Based on Collapse Analysis“
Writen by Zhaodong Shi, Yan Liang, and Jialei Yan
It is interesting paper, it is generally well written and fits in the field of an international journal of Applied Sciences. Therefore, this paper is recomended for publication in MDPI Applied Sciences journal with the suggested minnor corrections.
Remarks
Line 20, 22 Working condition 2 is mentioned in the abstract, it is not clear what this condition and what it mean?
The formulas should be reviewed and edited. The characters in the formulas and in the description must match. Character sizes must be checked (line 199, line 219 and so on). All characters used in the formulas must be described (Eq. 18, 15).
Line 262 It is not clear what does it mean parameter Sy?
There is a lack of explanations and indications as to why such values of coefficients have been dropped. For example line 122 t1=12,5 (dimensions?), line 123 D=12,5(dimensions); line 150, line 164, line 272 and so on.
Line 266 CEB-FIP Model 90 [35] is not mentioned in list of references. [35] is not this reference.
Line 274 …, The must be lower case letter.
Line 308 it is not clear witch above article you mean?
Table 2 how did you get the values of damage index? Why different groups of damage indices vary within such limits.
Table 3 The regression parameter R2 does not described?
In the manuscript and reference list does not mentioned your work
Yan Liang, Jia-lei Yan, Jun-lei Wang, Peng Zhang, Bao-jie He, "Analysis on the Time-Varying Fragility of Offshore Concrete Bridge", Complexity, vol. 2019, Article ID 2739212, 22 pages, 2019. https://doi.org/10.1155/2019/2739212
In this article are presented a part research of this manuscript.
Reviewer 3 Report
My comments are contained in the attached file: "ApplSci-1004925_REVIEW.pdf"

Reviewer 4 Report
OVERVIEW
The time-variant fragility of the four-span offshore continuous beam bridge has been analyzed, by means of the fragility analysis. The effect of the material deterioration on the seismic performance of the RC bridge has been investigated, considering the influence of concrete carbonation and chloride-induced corrosion. The collapse conditions have been studied, and quantification method for evaluating the fragility coefficient has been proposed.
GENERAL RECOMMENDATIONS
Dear Authors,
The research is interesting and novel.
In my opinion, the conducted studies are valuable and useful, however they should be presented clearly. The text is cumbersome and confusing and is not appropriate for publication in the present form, because it is difficult to read and to understand it. So, the spelling and grammar check is required.
That is why I recommend acceptance after major revisions.
Kind regards.
- GENERAL COMMENTS
1.1. The test and links must always be clear, e.g.:
line 308
“Based on [36]” instead of „Based on the above article”
1.2. Why the accepted service period achieves 150 years?
1.3. Have any verification techniques been used, perhaps, formal verification, model checking, probabilistic model checking (Markov chains or Markov decision processes)?
1.4. It would be useful to present codes and standards in the end of Chapter 1 for justification of the chosen study methodology.
- SELF-PLAGIARISM
The data of at least one previous paper of the Authors are used in this manuscript.
Figure 11. is identical to the Figure 11. in the previous paper of the Co-Authors https://doi.org/10.1155/2019/2739212
2.1. So, the Authors should give a reference to the previous publication. It is recommended not to copy the same figures and text in this manuscript or at least mention that the structure presented in Figure 3 has already been presented in the previous paper (even if this study is the continuation of the previous project).
2.2. If there is a strong need in pasting the same data, figures (instead of providing a reference), this fact should be mentioned in the Publication Ethics Statement according to the Journal rules and standards, and the agreement for copying the same data must be obtained from the Editors of the Journals where those data have already been published. Otherwise, it can be considered as self-plagiarism.
2.3. The numerical model has already been elaborated at least two years ago, so using the same data and figures for its detailed description in this paper, presenting particularly the Figure 11 as it has been just prepared in terms of the new research project, seems to be inappropriate.
2.4. So, the Authors might either avoid repetition or preparing the appropriate explanation.
It concerns both mentioned figure and all other data which have already been published (even if those papers are not Open Access and are unavailable for the Reviewers).
- ENGLISH
3.1. Careful editing of the English language and style all over the manuscript is strongly required. It is recommended to ask an English speaker for assistance.
E.g. lines 155-159:
The actual engineering has better performance due to the repair of concrete after cracking, so the actual corrosion rate and material properties of the reinforced concrete structure have a large error with the calculated results [25], to make the simulation results more accurate, in this paper, the repair of the damaged concrete pier is considered in the finite element model, as shown in Equation (21) – Equation (24) [25].
This sentence does not make sense. The structure is absolutely wrong.
There are numerous sentence construction errors, e.g. beginning with p.2, lines 82-89.
Concrete carbonation refers to the reaction of CO2 in the environment with the Ca(OH)2 in the 83 cement paste, to reduce its alkalinity and destroy the passivation film on the reinforcement bar surface, lead to reinforcement bar corrosion and concrete cracking, and finally cause durability damage [3-5].
This sentence is not clear. At least, it must be simplified and divided into 2-3.
At present, the carbonation depth is usually taken as the influence parameter in the research of concrete carbonation performance around the world [5].
The sentence is cumbersome.
In this case, and (???) the carbonation rate (S) is selected as the influence factor of concrete carbonation, and the effect of carbonation depth and section size is considered, as shown in Equation (1) – Equation (5) [22].
The keywords are missed, the conjunctions are overused.
And so forth.
3.2. The meanings of words are often misunderstood, and the sense is often missed. Sentences are often incomplete due to lack of keywords, verbs. The words should not be missed, the subject-verb agreement is always needed.
For example:
- lines 38-41 – two sentences are not clear, the verbs are missed
- the wrong forms of verbs is often used, which leads to difficulties in perceiving the text (e.g. line 51: “is applied to” instead of “is applies to”), etc.
- line 60 – “the method comprised” or “the authors considered” instead of “the method considered”
- line 70 – it is better to simplify: “component fragility research” instead of “research on the fragility of the component”
etc.
3.3. Abstract and Conclusions must be rewritten and carefully checked for correct grammar.
- REFERENCES
Most of the references of last years must have the DOI links, however, the Authors omitted them. It is recommended to provide the DOI links, wherever possible.
Round 2
Reviewer 4 Report
Dear Authors,
I am pleased to recommend this paper for publication.
Kind regards